# Diagnostic performance of eNose technology in detecting colorectal cancer recurrence: A prospective evaluation

Ivonne J. H. Schoenaker[1,2]*, Henderik L. van Westreenen[3], Evelyn J. Finnema[2], Ruud Schrauwen[4], Richard M. Brohet[5], Wouter H. de Vos Tot Nederveen Cappel[6]

**1** Oncology Center Isala, Zwolle, the Netherlands, **2** Department of Health Science, Nursing Research, University of Groningen, University Medical Center Groningen, Groningen, the Netherlands, **3** Department of Surgery, Zwolle, the Netherlands, **4** Department of Gastroenterology and Hepatology, Uden, the Netherlands, **5** Department of Epidemiology & Statistics, Zwolle, the Netherlands, **6** Department of Gastroenterology and Hepatology, Zwolle, the Netherlands

* i.j.h.schoenaker@isala.nl

## Abstract

### Introduction

After curative treatment for colorectal cancer (CRC), there is a 15% risk of recurrence. Early detection of an asymptomatic recurrence may lead to curative treatment options. To date, follow-up strategies do not have optimal sensitivity and specificity. In this prospective study, we aimed to assess the diagnostic performance of eNose technology to detect recurrent CRC following curative surgery.

### Materials and methods

A prospective evaluation study was performed to investigate whether eNose can discriminate patients with recurrent CRC following curative resection from patients without recurrent CRC based on VOC patterns during follow-up. The primary outcome measure is the diagnostic accuracy of eNose for detecting recurrence in CRC patients. With machine learning, a model was developed, and several performance metrics were used to evaluate the diagnostic performance of the eNose model.

### Results

A total of 406 patients who underwent curative resection for CRC between 2018−2023, were included in the study. VOC analysis was used to detect recurrent CRC during follow-up. Although the eNose model demonstrated promising results in the train set with an AUC of 0.90 (95% CI:0.84–0.96), the corresponding accuracy of 0.56 was low. Moreover, with a corresponding sensitivity of 0.52, accuracy of 0.51, and AUC of 0.51 (95% CI:0.38–0.64), the performances in the test set, declined.

**Data availability statement:** All relevant data are within the manuscript and its Supporting Information files.

**Funding:** The author(s) received no specific funding for this work.

**Competing interests:** The authors have declared that no competing interests exist.

**Abbreviations:** CRC, colorectal cancer; CEA, Carcinoembryonic Antigen; VOC, Volatile Organic Compounds; eNose, electronic nose; CI, Confidence Interval; AJCC, American Joint Committee on Cancer; BMI, Body Mass Index; PPV, Positive predictive value; NPV, Negative predictive value; AUC, Area under the curve; ROC, curve receiver-operating characteristics curve (ROC-curve).

## Conclusion

eNose technology is not able to accurately detect recurrent CRC after curative resection of the primary tumour. Larger studies are needed before clinical implementation can be realized, while the lack of reproducibility must be addressed.

## Introduction

Following curative treatment for colorectal cancer (CRC), approximately 15–30% of patients develop a recurrence [1–3]. Detecting asymptomatic recurrences at an early stage is crucial to enable potential curative treatment. Therefore, patients undergo follow-up monitoring. Most recurrences occur within the first two years post-treatment and are detected by imaging or assessments of the tumour marker Carcinoembryonic Antigen (CEA).

In the Netherlands, follow-up guidelines for CRC stages II and III have transitioned to a CEA-triggered approach [4]. This protocol includes regular CEA assessments for up to five years post-surgery, while imaging is limited to a single CT scan one year after surgery. Limited sensitivity hinders CEA testing's effectiveness. [3,5]. for instance, using CEA, Kievit et al. demonstrated a sensitivity of 72% for detecting liver metastases and 60% for locoregional recurrences [6]. Additionally, CEA is prone to be false-positive due to inflammatory diseases, smoking, or other cancers (e.g., pancreatic and lung). These limitations justify the search for new diagnostics modalities for detecting recurrent disease.

The analysis of volatile organic compounds (VOCs) in exhaled air is gaining interest as a non-invasive diagnostic tool for CRC detection. Each individual has a unique 'breath print', which reflects their health status and consists of VOCs, gaseous products of metabolism [7]. VOCs can serve as biomarkers for diseases, including several cancers [8–11]. In CRC, metabolic and microbiome alterations produce distinct VOC profiles that can be measured in exhaled air. Breath analysis of VOCs has also shown potential in detecting CRC recurrence. Markar et al. demonstrated reduced propanal levels following surgery that subsequently increased with recurrence, showing a sensitivity of 71% and a specificity of 91% [7,12].

Electronic nose (eNose) devices that utilize pattern-recognition techniques to analyse VOC profiles offer a non-invasive approach to detecting CRC. Van Keulen et al. reported diagnostics accuracies of 0.73 and 0.84 in differentiating CRC patients from advanced adenomas and healthy individuals, respectively [8]. A systematic review and meta-analysis of Wang et al. reported pooled sensitivity and specificity of 0.88 and 0.85 for CRC detection via VOC analysis and 0.87 and 0.78 specifically for eNoses [13]. However, van Riswijk er al. recently evaluated eNose for CRC detection in FIT-positive patients, along with reproducibility and external validation. The eNose showed poor diagnostic performance (AUC 0.54; sensitivity 0.39, specificity 0.68) and low reproducibility (ICC 0.22) [14].

VOC pattern alteration following curative surgery for CRC has been demonstrated with an accuracy of 0.75 [15]. Earlier studies also showed eNose's ability to detect

CRC recurrence with a diagnostic accuracy of 0.81 [13]. If eNose technology can accurately detect CRC recurrence, it could represent a minimally invasive, patient-friendly diagnostic tool with real-time results. A positive breath test could serve as an indicator for additional diagnostic work to localize and stage recurrent CRC.

In a pilot study of our group, the feasibility of eNose in identifying extra luminal local recurrence or metastases of CRC was studied. This study showed a sensitivity and specificity of 0.88 and 0.75, respectively, with an overall accuracy of 0.81 [16].

This prospective evaluation study aimed to assess the diagnostic performance of eNose technology in detecting CRC recurrence among patients who had undergone curative resection of the primary tumour.

## Materials and methods

### Study design and patient selection

This prospective study was conducted in Isala Oncology Center, Zwolle, the Netherlands. Between January 1, 2020, and April 14, 2024, patients with stage I-III, or stage IV with synchronous or metachronous metastases treated with curative intent, who underwent surgery from January 2018 to June 2023, were asked to participate. Exclusion criteria were inability to perform the breath test, insufficient understanding of the Dutch language, local resection of the tumour and another malignancy in the past five years (except for basal-cell carcinoma). Patient inclusion started in January 2020, but the first breath tests were performed in August 2020 due to the COVID-19 pandemic. The Medical Ethics Review Committee (METC) of Isala, Zwolle, declared that the study protocol is not subject to the Medical Research Involving Human Subjects Act (WMO) (Isala METC 190110). The study was registered in the Dutch trial registration (NL9084). Written informed consent was obtained from all participants before inclusion.

### Follow-up data collection

During the study period patients who were followed up after a curative resection of CRC underwent imaging and CEA assessments according to the applicable guideline of 2020. Imaging consisted of an ultrasound of the liver and X-thorax. Breath test outcomes were compared with regular controls to determine the presence or absence of recurrent CRC. Histological or cytological confirmation was not obligatory, as this was not always possible or desirable. The gold standard for recurrent CRC was the determination during the multidisciplinary team meeting based on at least one CT scan. Breath tests were scheduled alongside routine follow-up visits during the first three years, or later in case of abnormal findings and suspected recurrence. Depending on patient willingness, multiple tests were performed during follow-up, resulting in a total of 944 breath tests across 406 patients. For analysis, either the first successful breath test or the test at recurrence was used.

Before breath samples were taken, exogenous factors (e.g., smoking, medication, alcohol, fasting) and endogenous patient characteristics (e.g., Body Mass Index (BMI) or specific comorbidities) that might influence the VOC composition were collected [17]. The use of supplements was added as a variable due to the high prevalence of supplement use among cancer patients [18]. Discomfort during the test was assessed on a 0–10 scale.

Patients breathed into the device for five minutes through a disposable mouthpiece provided with carbon filters to prevent contamination of the inhaled air with environmental VOCs. All patients wore a nose clip and were instructed to close their lips firmly around the mouthpiece to avoid pollution with unfiltered air. The eNose Company trained healthcare practitioners to standardize the breath test.

### Aeonose™ technology and model development

The primary endpoint was the accuracy of eNose in detecting recurrent CRC during follow up. Breath tests were conducted with two CE-certified Aeonose™ devices from the eNose Company (Zutphen, The Netherlands). From

September 2023, only the newer device was used, on the eNose Company's request. The Aeonose™ technology has successfully been used and described for lung cancer diagnosis [19]. The Aeonose™ contains three micro hotplate metal-oxide sensors that behave like semiconductors. These sensors contain various types of metal and catalysing agents. The present VOCs in the exhaled breath provoke a redox reaction on the surface of the sensors that subsequently changes the measured conductivity. The redox reactions are dependent on the present VOCs, types of sensors, reaction dynamics, and temperature. The Aeonose™ uses thermal cycling, in which the temperature varies between 260 and 320°C, thus allowing the generation of specific VOC signals. Recording the passing of this thermal cycle with each specific sensor obtains a specific and unique pattern that resembles the measured gas composition. One single breath test generates a data matrix of 64x36 conductivity values per sensor. After preprocessing, the data are compressed using singular value decomposition to avoid overfitting. Compression of the data then generate one vector with a length of 17. This single vector is used as input to train the different machine learning algorithms and to classify them as accurately as possible [15]. The performance of different machine-learning models was evaluated by the proprietary software program 'Aethena' version 2.64.

Breath data were divided into training (76%) and test (24%) data set. The training set was used to train and develop the model including K-fold cross-validation technique to avoid overfitting. The test set was used for internal validation of the trained algorithm. Breath data were randomly assigned to the training and the test set, ensuring at least 25 patients with recurrences in both sets.

## Model performance

Model performance was evaluated including the following performance metrics sensitivity, specificity, accuracy, positive predictive value (PPV), negative predictive value (NPV), and the area under the curve (AUC) of the 'receiver-operating characteristics curve (ROC-curve)'. We expected the eNose to achieve a sensitivity of at least 70% for the detection of recurrence, similar to standard CEA testing. We expected the sensitivity would improve with the potential of the self-learning ability of machine learning. Model performance was evaluated, with the optimal model determined using a Random Forest algorithm. A threshold of −0.63, provided the best separation between the two groups, optimizing sensitivity and specificity.

## Statistical analysis

The primary endpoint is the accuracy of eNose in detecting recurrent CRC. Patient characteristics were summarized by count and proportion for categorical data, by mean and standard deviation for normally distributed continuous data or median and interquartile range for non-normal distributed data. T-tests, Mann–Whitney U test, Chi-squared ($X^2$), or Fishers exact test were applied as appropriate to assess differences between the groups. A (two-sided) p-value < 0.05 was considered significant. All analyses were performed using Statistical Package of Social Sciences version 24.0 (SPSS, IBM, Armonk, NY, USA).

## Results

### Study population

Between January 2020 and April 2024, 498 out of 748 eligible patients provided informed consent and were included in the study, during which breath tests were performed. Ninety-two patients were excluded due to a failed breath test (n = 38), changes in health status (n = 30), technical issues (n = 19), wrong inclusion (n = 3), or loss to follow up (n = 2). A total of 406 patients completed breath tests suitable for analysis. Among these, 63 patients (15%) developed a recurrence. The mean age of the study population was 68 years (SD 11), with a range 29–90 years. Forty-one percent were female. Baseline patient, tumour and breath test characteristics are presented in Table 1.

**Table 1. Baseline patient, tumour and breath-test characteristics.**

| | Total | No recurrence | Recurrence | P-value |
|---|---|---|---|---|
| | n = 406 | n = 343 | n = 63 | |
| **Patient characteristics** | | | | |
| **Age** | | | | 0.584[c] |
| Mean ± SD | 68 (11) | 68 (11) | 68 (13) | |
| **Gender** | | | | 0.06[a] |
| Female | 166 (41) | 147 (43) | 19 (30) | |
| **BMI kg/m2** | | | | 0.398[c] |
| Mean ± SD | 26,6 (4,5) | 26,7 (4,6) | 26,2 (3,8) | |
| **ASA** | | | | 0.713[a] |
| I | 106 (26) | 87 (25) | 19 (30) | |
| II | 256 (63) | 218 (64) | 38 (60) | |
| ≥ III | 44 (11) | 38 (11) | 6 (9) | |
| **Comorbidity** | | | | 0.224[a] |
| Yes | 284 (70) | 244 (71) | 40 (63) | |
| **Tumour characteristics** | **Total** | **No recurrence** | **Recurrence** | **P-value** |
| **Localization primary tumour** | | | | 0.108[a] |
| RCC | 167 (41) | 147 (43) | 20 (32) | |
| LCC | 137 (34) | 116 (34) | 21 (33) | |
| Rectal | 102 (25) | 80 (23) | 22 (35) | |
| **MMR-status** | | | | **0.020[b]** |
| MMRp (proficient | 247 (61) | 202 (59) | 45 (71) | |
| MMRd (deficient) | 41 (10) | 40 (12) | 1 (2) | |
| Missing | 118 (29) | 101 (29) | 17 (27) | |
| **Neo-adjuvant therapy** | | | | **0.009[a]** |
| Yes | 80 (20) | 60 (17) | 20 (32) | |
| **Tumour Stage**\*\* | | | | **0.000[b]** |
| I | 65 (16) | 64 (19) | 1 (2) | |
| II | 126 (31) | 118 (34) | 8 (13) | |
| III | 187 (46) | 150 (44) | 37 (59) | |
| IV | 28 (7) | 11 (3) | 17 (27) | |
| **Adjuvant therapy** | | | | 0.085[a] |
| Yes | 112 (28) | 89 (26) | 23 (37) | |
| **Breath test characteristics** | **Total** | **No recurrence** | **Recurrence** | **P-value** |
| **Time after surgery (months)** | | | | **0.041[d]** |
| Mean ± SD | 17 (14) | 16 (12) | 22 (20) | |
| Median (IQR) | 11 (16) | 11 (16) | 14 (28) | |
| **eNose device** | | | | **0.000[a]** |
| Nr 40, older device | 324 (80) | 286 (83) | 38 (60) | |
| Nr 13, newer device | 82 (20) | 57 (17) | 25 (40) | |
| **Current smoking** | | | | 0.100[a] |
| Yes | 31 (8) | 23 (7) | 8 (13) | |
| **Diet** | | | | 1.000[b] |
| Yes | 29 (7) | 25 (7) | 4 (6) | |
| **Last meal** | | | | 0.095[a] |
| < 3 hours | 144 (36) | 116 (34) | 28 (45) | |
| >3 Hours | 258 (64) | 224 (66) | 34 (55) | |

*(Continued)*

**Table 1.** (Continued)

| | Total | No recurrence | Recurrence | P-value |
|---|---|---|---|---|
| | n = 406 | n = 343 | n = 63 | |
| **Alcohol < 24 hours (missing 36)** | | | | 0.151[a] |
| Yes | 122 (33) | 107 (35) | 15 (25) | |
| **Stoma** | | | | 0.117[a] |
| Yes | 58 (14) | 45 (13) | 13 (21) | |
| **Medication** | | | | 0.988[a] |
| Yes | 271 (67) | 229 (67) | 42 (67) | |
| **Supplements** | | | | 0.240[a] |
| Yes | 157 (39) | 137 (40) | 20 (32) | |
| **CEA serum level ng/ml** | | | | **0.000[a]** |
| <5 ng/ml | 343 (85) | 315 (93) | 28 (46) | |
| ≥ 5 ng/ml | 58 (15) | 25 (7) | 33 (54) | |

Data are expressed as n (%) unless otherwise specified.

** in case of a ypTNM stage the cTNM classification is taken.

ASA;American Society of Anesthesiologists; BMI; Body mass index; RCC; coecum to splenic flexure, LCC; splenic flexure to rectum, included recto-sigmoid, RC; rectal cancer; MMR; mismatch repair.

[a]Pearson Chi-Square test, [b]Fisher's Exact Test, [c]Independent-Samples T-test, [d]Mann-Whitney U test.

## Tumour and recurrence characteristics

The prevalence of recurrence was 15%. However, prevalence differed between the training (11%) and test set (25%). Seventy percent of the recurrences were in a single organ, with the liver (39%) lungs (27%) and locoregional (20%) as the most common localizations. A multidisciplinary board confirmed all recurrences and 57% were pathologically confirmed. As expected, the primary tumour stage was more advanced among those who had recurrent CRC. Those with recurrence had more often stage III and IV CRC (p < 0.001) and a higher proportion of Mismatch Repair proficient (MMRp) tumours, 71% versus 59% (p = 0.020). Patients with a recurrence were more male (70%), however this difference was not statistically significant.

Significant differences in breath test characteristics between recurrence or no recurrence include greater use of the newer device among recurrence cases (40% vs 17%, p < 0.001) and a longer median interval between surgery and breath testing (14 vs 11 months, p = 0.041).

## Breath test characteristics

The majority of patients (60%) underwent multiple breath tests, ranging from one to five. In total, 944 breath tests were performed, of which 851 (90%) were successful. Breath test failures were attributed to patient-related factors (e.g., dyspnoea or panic, n = 40) or technical issues with the eNose device or internet/WIFI connectivity (n = 53). Patient inconvenience, as measured on a Numeric Rating Scale (0–10) had a mean score of 1.73 (SD 1.85) and a median of one, with zero representing no inconvenience. Ninety-seven percent of all patients expressed willingness to perform future breath tests. All breath tests were performed within the same outpatient clinic.

## Model performance

There were no significant differences in baseline characteristics between patients in the training and test set, presented in S1 Table. The training set included 307 patients, of whom 38 (11%) experienced disease recurrence. The model

demonstrated the ability to discriminate between patients with and without recurrence, achieving a sensitivity of 0.92 (95% CI; 0.83–1.0) and a specificity of 0.52 (95% CI; 0.46–0.58). The accuracy was 0.56 (95% CI; 0.51–0.62) and the AUC of the ROC curve was 0.90 (95% CI; 0.84–0.96). The data analysed is available in S1 Dataset.

The test set included 99 patients, 25 (25%) of whom experienced recurrence. There were 13 true positives, 37 false positives, 12 false negatives and 37 true negatives in the classified breath tests. The model demonstrated a sensitivity of 0.52 (95%CI; 0.32–0.71) and a specificity of 0.50 (95% CI; 0.39–0.61) in detecting recurrences. The accuracy was 0.51 (95%CI; 0.41–0.60) and the AUC of the ROC curve 0.51 (95% CI;0.38–0.64) (Table 2). Lowering the threshold to −0.70 increases sensitivity to 0.60 but reduces accuracy to 0.49. We selected the threshold that provided the highest accuracy.

No significant differences were identified between correctly and incorrectly predicted recurrences, except for using supplements. The incorrectly predicted group had a higher prevalence of supplement use, including vitamins, minerals, and herbs (50% vs 27%; p = 0.017). Patient, tumour and breath test characteristics for correctly and incorrectly predicted patients in the test set are presented in S2 Table.

Patients without recurrence (n = 343) were followed up at least six months after the breath to verify their clinical status. Seven patients developed recurrence during follow-up. At the time of recurrence, no breath test was available for these patients, either because they declined performing a breath test at that moment, or because recurrence was not detected in scheduled follow-up and therefore no breath test was planned. Their initial breath test result had been positive, while routine clinical follow-up at that time showed no signs of recurrence. These cases were therefore classified as false positives.

## Discussion

This prospective study assessed the diagnostic performance of eNose technology for detecting recurrent colorectal cancer (CRC) via volatile organic compound (VOC) analysis in exhaled breath. Ideally, a well-fitted model performs consistently across both training and test sets, indicating generalizability. Our training set showed promising sensitivity (0.92), but low accuracy (0.56). In contrast, the test set revealed a marked decline in all performance metrics (sensitivity 0.52, accuracy 0.51), suggesting the eNose currently lacks clinical utility for detecting recurrent CRC

At the start of our study, Van Keulen had published promising results on VOC-based CRC detection, and we participated in an external validation study [8,14]. However, their recent findings showed poor predictive performance of the eNose (AUC 0.54). Although earlier studies reported high sensitivity (0.87–0.93) and specificity (0.78–0.89), most were feasibility studies with limited sample sizes [13,20]. For example, Wang et al.'s review noted that only Van Keulen's study included a larger cohort (n = 447), with 70 CRC cases [13]. Their reported sensitivity (0.95) and specificity (0.64) closely

**Table 2. Performance of eNose detection model 95% CI.**

|  | Training set | Test set |
|---|---|---|
|  | N = 307 | n = 99 |
| **Prevalence recurrence** | 11% | 25% |
| **Sensitivity** | 0.92 (0.83-1.0) | 0.52 (0.32–0.71) |
| **Specificity** | 0.52 (0.46-0.58) | 0.50 (0.39–0.61) |
| **PPV** | 0.21 (0.15–0.27) | 0.26 (0.14–0.38) |
| **NPV** | 0.98 (0.95–1.0) | 0.75 (0.63–0.87) |
| **Accuracy** | 0.56 (0.51–0.62) | 0.51 (0.41–0.60) |
| **AUC** | 0.90 (0.84–0.96) | 0.51 (0.38–0.64) |

*The threshold in the test set was set on −0.63. PPV; Positive Predictive Value, NPV; Negative Predictive Value, AUC; Area under the curve.*

match our training set results, though accuracy was not provided. Overall, these findings underscore the need for larger studies to determine whether VOC analysis can reliably detect CRC recurrence.

Several factors may explain the low performance observed, including overfitting, limited robustness, endogenous and exogenous influencing factors, and technical limitations. Overfitting, where a model performs well on training data but poorly on unseen data, was evident [21,22]. Despite applying K-fold cross-validation, robustness remained limited, likely due to small sample size and low recurrence prevalence. Although the overall prevalence matched the anticipated 15%, the imbalance between training (11%) and test sets (25%) may have biased performance comparisons. Moreover, 15% recurrence may still be insufficient for reliable detection.

Higher prevalence appears to improve model performance. For instance, Kort et al.'s lung cancer screening study, with prevalence rates of 40–43%, achieved an AUC of 0.87 [19]. As van Riswijk et al. noted, low prevalence can impair model accuracy [14]. Developing a robust and generalizable model likely requires thousands of training examples, challenging in the context of recurrent CRC [14].

Diagnostic accuracy of eNose technology varies widely across studies. Multiple endogenous and exogenous factors can influence VOC composition, affect breath profiles and potentially confound diagnostic outcomes. Variables such as smoking, comorbidities, diet, age, sex, BMI, and medication have been identified; however, findings in the literature remain contradictory [23–25]. The sensitivity of electronic noses to these variations contributes to inconsistencies in diagnostic accuracy across different patient populations and research settings [20].

In our study, supplement use was the only significant difference, with a higher prevalence among incorrectly classified patients (50% vs. 27%). Although direct evidence linking supplements to VOC composition is lacking, some research suggests that specific nutrients, such as vitamins, herbs, and minerals, may impact gut microbiota, digestion, and overall health, indirectly influencing breath composition [24,26]. Given the wide range of factors that can alter VOC profiles, controlling for these potentially confounders remain a challenge in VOC-based diagnostics, requiring careful study design.

One inherent technical limitation of e-noses is sensor drift—a gradual change in sensor output independent of sample composition. This drift may reduce instrument sensitivity over time, increasing the likelihood of false diagnoses [20]. During our study, we discontinued use of an older eNose device upon recommendation from "The eNose Company". In the test set, incorrect predictions were more frequent with the older device (53% vs. 37%), although this difference was not statistically significant. These findings highlight usability concerns that warrant further investigation.

Strengths of this study include its prospective design, homogeneous consecutive patient cohort which closely reflects a real-world population, and standardized follow-up protocol. We also systematically collected data on potential confounders, enhancing the robustness of our findings. Previous studies often failed to report such variables [20]. While our sample size met the target prevalence, the single-centre design limited recruitment and generalizability.

This study illustrates both the promise and limitations of eNose technology for detecting recurrent CRC. To improve model robustness and diagnostic accuracy, future research should focus on larger, multicentre trials with higher recurrence prevalence. However, the inherently low recurrence rate poses challenges for assembling sufficiently large training datasets [21]. Moreover, systematic reporting of confounding factors is essential to enhance generalizability and ensure reliable performance across clinical settings.

Despite its potential as a non-invasive diagnostic tool, current eNose technology is not yet suitable for detecting recurrent CRC in clinical practice

## Conclusion

eNose technology is currently unable to detect recurrent CRC with sufficient accuracy and has no clinical utility. While the training set showed high sensitivity, test set results revealed issues related to overfitting, data variability, and technical limitations. Further research is needed to overcome these challenges and establish the role of VOC-based diagnostics in CRC surveillance.

## Supporting information

**S1 Table. Patient, tumour and breath test characteristics for the training and test set.**
(DOCX)

**S1 Dataset. Results breath test used for performance analysis.**
(XLSX)

**S2 Table. Patient, tumour and breath test characteristics for the correctly and incorrectly predicted patients in the test set.**
(DOCX)

## Acknowledgments

We thank colleagues, nurses and medical assistants for administering the breath tests. We also thank The eNose Company, Zutphen, The Netherlands, for supplying the Aeonose™ devices, including software packages, filters and mouthpieces. Informed consent was obtained from all subjects involved in the study.

## Author contributions

**Conceptualization:** Ivonne J.H. Schoenaker, Henderik L. van Westreenen, Richard M. Brohet, Wouter H. de Vos Tot Nederveen Cappel.

**Formal analysis:** Ivonne J.H. Schoenaker, Henderik L. van Westreenen, Richard M. Brohet, Wouter H. de Vos Tot Nederveen Cappel.

**Methodology:** Ivonne J.H. Schoenaker, Henderik L. van Westreenen, Richard M. Brohet, Wouter H. de Vos Tot Nederveen Cappel.

**Supervision:** Wouter H. de Vos Tot Nederveen Cappel.

**Validation:** Ivonne J.H. Schoenaker, Henderik L. van Westreenen, Wouter H. de Vos Tot Nederveen Cappel.

**Visualization:** Ivonne J.H. Schoenaker, Wouter H. de Vos Tot Nederveen Cappel.

**Writing – original draft:** Ivonne J.H. Schoenaker.

**Writing – review & editing:** Ivonne J.H. Schoenaker, Henderik L. van Westreenen, Evelyn J. Finnema, Ruud Schrauwen, Richard M. Brohet, Wouter H. de Vos Tot Nederveen Cappel.

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
