## [Decision Letter · Decision Letter 0]

13 Oct 2025

Dear Dr. schoenaker,

Thank you for submitting your manuscript to PLOS ONE. After careful consideration, we feel that it has merit but does not fully meet PLOS ONE’s publication criteria as it currently stands. Therefore, we invite you to submit a revised version of the manuscript that addresses the points raised during the review process.

We look forward to receiving your revised manuscript.

Kind regards,

Rajeev Singh

Academic Editor

PLOS ONE

Journal Requirements:

Reviewers' comments:

Reviewer's Responses to Questions

**Comments to the Author**

1. Is the manuscript technically sound, and do the data support the conclusions?

Reviewer #1: Partly

2. Has the statistical analysis been performed appropriately and rigorously?

Reviewer #1: I Don't Know

3. Have the authors made all data underlying the findings in their manuscript fully available?

Reviewer #1: Yes

4. Is the manuscript presented in an intelligible fashion and written in standard English?

Reviewer #1: Yes

Reviewer #1: Dear authors,

I have read with interest your manuscript entitled "Diagnostic performance of eNose technology in detecting colorectal cancer recurrence: a prospective evaluation”.

The manuscript describes the validation of a previously developed eNose model, used to detect extraluminal locoregional recurrences or metastases in CRC.

Strength:

Validation was performed in a prospectively collected cohort, including a relatively large number of patients and samples.

Main concern:

In its current form, the manuscript lacks detail regarding the model development. In its current form, it seems as if model development was done in your previous study, with validation being the purpose of the current study. If so, I can see that the model development has been extensively described in your previous study, but merely providing just a bit more information than a reference is not enough, specifically since the training set forms an integral part of the model development.

The manuscript is confusing; it seems as if the available data in this study were divided in a training and a testing set. However, the purpose of this study was to validate the model, and from the text and the results of the previous study, it seems as if the previous study was used as the training set, and the results of the current study as validation and test sets (?). If so, the training set that was used, was a lot smaller than the validation and test sets, which should not be the case. If not (and the model was trained in the current study), this should be explained in more detail and a validation set should be added.

Specific comments:

Abstract:

The M&M section is not completely correct; this section should describe what was done in this study, not include M&M of a previous study

Introduction:

- “Limited sensitivity…, false negative results.” The second part of the sentence is superfluous, please consider removing for conciseness.

- I would suggest focusing more on the fact that the current study concerns a continuation of the previously published study, namely the creation of the validation and test sets. The coherence of this study with the previous study could be more clear.

M&M

- just providing confirmation that patients provided written informed consent might suffice.

- at what time point were patients selected? Directly after surgery, or at varying post-therapeutic intervals?

-were patients excluded if they received chemotherapy or immunotherapy in the past three months? (exclusion criterion in the pilot study)

- When were breath control results compared with regular controls to determine presence or absence of recurrent CRC? If compared with the gold standard at the end of the study, this should not have to be mentioned in the text here. If results were compared with regular results during the study, why was this done? And what was done with divergent results?

- “The first successful breath test, or …. was used”. Why were not all breath results used?

- I find the alinea called “Model development” confusing as is. The model has already been developed in the previous study, while ongoing development and validation of the model is done in this study, or so it seems from the text. Please consider being more precise. Also, the labelling of the different sets is confusing: basically, models should be fitted on a training data set, with a validation set for unbiased evaluation of the model, and a test set for final evaluation of the model.

- wat was the rationale for the division between the training and the test set? And if there was indeed a “real” training set in this study, where is the validation set?

Results

- why were only 498 of the 748 eligible patients included?

- please describe how many breath tests were performed with the newer eNose and the older eNose (from table 1 it does not become clear whether device No 40 or No 13 is the newer device).

- a greater number of recurrences was seen using the newer device, please provide the number of uses versus the number of recurrences in the text.

- at the time of FU 6 months after the breath test, 7 patients had developed a recurrence. Why were 4 of these labelled as false positive? Specifically since a follow-up period of 6 months after the last breath test was mandatory to verify the results of the regular follow-up?

- please provide the raw data for the training set as well (raw data are provided for the test set)

- Please consider providing more insight (raw data) as to how exogenous factors and endogenous patient characteristics differed between the eNose categories: in the text now it is only mentioned that there were no significant differences with the exception of supplement use.

Discussion

- please consider leaving out the paragraph citing Markar as these results concern VOC-analysis by GC-MS whereas in this study VOC-analysis by eNose technology is studied.

- A target prevalence of 15% is mentioned in the discussion, please provide information about this percentage in the M&M section.

- why was the training set (pilot study) not mentioned in the discussion? The training set consisted of a relatively low number of samples (66 patients). In machine learning, the training set should be (al lot) larger than the validation and test sets.

**Do you want your identity to be public for this peer review?** For information about this choice, including consent withdrawal, please see our Privacy Policy

Reviewer #1: No

---

## [Author Response · Author response to Decision Letter 1]

12 Nov 2025

At the request of the medical specialist, the conflict of interest paragraph has been reinstated. This was subject to some discussion within our research group, as the eNose company has ceased to exist following bankruptcy

---

## [Editor Report · Decision Letter 1]

18 Dec 2025

Diagnostic Performance of eNose Technology in Detecting Colorectal Cancer Recurrence: A Prospective Evaluation

PONE-D-25-45459R1

Dear Dr. schoenaker,

We’re pleased to inform you that your manuscript has been judged scientifically suitable for publication and will be formally accepted for publication once it meets all outstanding technical requirements.

Kind regards,

Rajeev Singh

Academic Editor

PLOS One
---

## [Editor Report · Acceptance letter]

PONE-D-25-45459R1

PLOS One

Dear Dr. Schoenaker,

I'm pleased to inform you that your manuscript has been deemed suitable for publication in PLOS One. Congratulations! Your manuscript is now being handed over to our production team.

Kind regards,

on behalf of

Dr. Rajeev Singh

Academic Editor

PLOS One